# Loss of circadian protection against influenza infection in adult mice exposed to hyperoxia as neonates

Yasmine Issah[1†], Amruta Naik[1†], Soon Y Tang[2], Kaitlyn Forrest[1], Thomas G Brooks[2], Nicholas Lahens[2], Katherine N Theken[2,3], Mara Mermigos[1], Amita Sehgal[4,5], George S Worthen[1,6], Garret A FitzGerald[2,3,4], Shaon Sengupta[1,2,4,6]*

[1]The Children's Hospital of Philadelphia, Philadelphia, United States; [2]Institute of Translational Medicine and Therapeutics (ITMAT), University of Pennsylvania, Philadelphia, United States; [3]Systems Pharmacology University of Pennsylvania Perelman School of Medicine, Philadelphia, United States; [4]Chronobiology and Sleep Institute, University of Pennsylvania, Philadelphia, United States; [5]Department of Neuroscience, University of Pennsylvania Perelman School of Medicine, Philadelphia, United States; [6]Department of Pediatrics, University of Pennsylvania Perelman School of Medicine, Philadelphia, United States

**Abstract** Adverse early-life exposures have a lasting negative impact on health. Neonatal hyperoxia that is a risk factor for bronchopulmonary dysplasia confers susceptibility to influenza A virus (IAV) infection later in life. Given our previous findings that the circadian clock protects against IAV, we asked if the long-term impact of neonatal hyperoxia vis-à-vis IAV infection includes circadian disruption. Here, we show that neonatal hyperoxia abolishes the clock-mediated time of day protection from IAV in mice, independent of viral burden through host tolerance pathways. We discovered that the lung intrinsic clock (and not the central or immune clocks) mediated this dysregulation. Loss of circadian protein, *Bmal1*, in alveolar type 2 (AT2) cells recapitulates the increased mortality, loss of temporal gating, and other key features of hyperoxia-exposed animals. Our data suggest a novel role for the circadian clock in AT2 cells in mediating long-term effects of early-life exposures to the lungs.

*For correspondence:
SenguptaS@email.chop.edu

†These authors contributed equally to this work

## Introduction

Hyperoxia represents the single most important toxic exposure to premature neonatal lungs and is the key risk factor for chronic lung disease of prematurity or bronchopulmonary dysplasia (BPD) (*Jobe and Bancalari, 2001*; *Peek et al., 2017*). Despite remarkable improvements in the survival of premature neonates, almost 40% of infants born at <29 weeks of gestation (approximately 10,000 new cases in USA each year) suffer from BPD (*Eunice Kennedy Shriver National Institute of Child Health and Human Development Neonatal Research Network et al., 2010*; *Eunice Kennedy Shriver National Institute of Child Health and Human Development Neonatal Research Network et al., 2015*). In surviving adults, BPD is associated with an increased risk of respiratory infections, such as asthma (*Gough et al., 2014*; *Yang et al., 2020*) and chronic obstructive pulmonary disease (COPD) (*Islam et al., 2015*; *McGrath-Morrow and Collaco, 2019*; *Savran and Ulrik, 2018*). As an increasing number of prematurely born neonates survive into adulthood, these long-term morbidities have acquired greater public health importance (*Jain, 2015*). In a murine model, neonatal hyperoxia increased morbidity from influenza A virus (IAV) (*O'Reilly et al., 2008*) in a dose-dependent manner (*Maduekwe et al., 2015*) through pathways involving both

alveolar type 2 (AT2) epithelial cells and immune pathways (*Buczynski et al., 2013*). Furthermore, this effect seems independent of the pathogen, since Yee et al. have recently demonstrated that the expression of the SARS-COV2 receptor, ACE2 in club cells, and AT2 cells is increased in adult mice exposed to hyperoxia as neonates (*Yee et al., 2020*).

We have previously demonstrated that circadian rhythms offer a protection against IAV, wherein mortality is threefold lower if the animals are infected in the morning than in the evening (*Sengupta et al., 2019*). This time of day-specific protection is lost upon genetic disruption of the clock via global deletion of core clock gene, *Bmal1*, which exacerbates immunopathology independent of viral replication. In keeping with their evolution imperative, circadian rhythms temporally segregate physiological process that allows organisms to anticipate its environment and adapt accordingly. The suprachiasmatic nucleus (SCN) is the master circadian pacemaker; however, peripheral tissues, including the lung, also have cell autonomous clocks (*Spengler and Shea, 2000*). Changes in oxygen tension are known to affect the clock (*Walton et al., 2018*; *Wu et al., 2017*). Low oxygen levels or hypoxia stabilize hypoxia-inducible factor (HIF). There exists a reciprocal relationship between the circadian clock and *Hif1a* mediated at the genomic level (*Peek et al., 2017*; *Wu et al., 2017*). More recently, even a short burst of hypoxia was shown to desynchronize the relationship between the SCN and peripheral clocks (*Manella et al., 2020*), which would confer further risk under conditions of stress. However, the effect of hyperoxia, especially very early in life, on circadian regulation of the recovered lung has not been investigated.

The early neonatal period represents a critical window for the development and consolidation of many important pathways, including circadian rhythms (*Bartman et al., 2020*; *Rivkees, 2007*; *Yang et al., 2014*). Early-life exposure to light (*Coleman and Canal, 2017*; *Smarr et al., 2017*), inflammation (*Adler et al., 2014*), or alcohol has disruptive effects on circadian rhythms in adulthood (*Allen et al., 2005*). But despite the high incidence of hyperoxia in premature neonates, how neonatal hyperoxia affects the development of circadian regulatory networks is not known. We hypothesized that early-life hyperoxia disrupts the development of circadian rhythms and that such disruption undermines recovery from lung injury. Here we test this hypothesis using an IAV infection model of adult mice exposed to neonatal hyperoxia. We find that indeed exposure to neonatal hyperoxia abrogates the time of day protection from circadian regulation of IAV infection through primarily host tolerance rather than anti-viral effects. This effect appears stage specific to the saccular stage of development that corresponds to the period during which the mice were exposed to hyperoxia in our model since adult mice exposed to hyperoxia did not lose the time of day-specific protection. To identify the location of the relevant clock, we used tissue-specific adult onset deletion of the core clock gene *Bmal1*. We report that disrupting *Bmal1* in AT2 cells of the lung faithfully recapitulates the phenotype of animals exposed to neonatal hyperoxia, suggesting that early-life hyperoxia disrupts the circadian regulation of the pulmonary response to hyperoxia through the AT2 clock.

## Results

### Neonatal hyperoxia abrogates the circadian protection from influenza infection in recovered adults

We previously showed that mice infected at ZT11 (ZT11 refers to 'Zeitgeber Time' 11 or dusk which marks the beginning of the active phase in mice since they are nocturnal) had threefold higher mortality than mice infected at ZT23 (dawn or at the beginning of their rest phase) (*Sengupta et al., 2019*). Upon global disruption of the clock (*Bmal1fl/fl:CAGGcreERt2/+*), in adult animals, the time of day difference was lost such that the mortality was high irrespective of the time of day of infection (*Sengupta et al., 2019*). Here we hypothesized that neonatal hyperoxia would disrupt circadian rhythms to result in a loss of temporal difference in outcomes in recovered adult mice infected with IAV at ZT11(dusk) or ZT23 (dawn). Mice exposed to either hyperoxia or room air as neonates were recovered to adulthood and infected with IAV (H1N1 mouse-adapted strain PR8) at either ZT23 or ZT11 (*Figure 1a*). Mice are born at the saccular stage of lung development that corresponds to 24–28 weeks gestation for humans (*Warburton et al., 2010*) and thus neonatal hyperoxia provides an excellent model to study BPD (*Hilgendorff et al., 2014*). The room air controls maintained the time of day difference in mortality; the group infected at ZT11 had a mortality 2.6 times

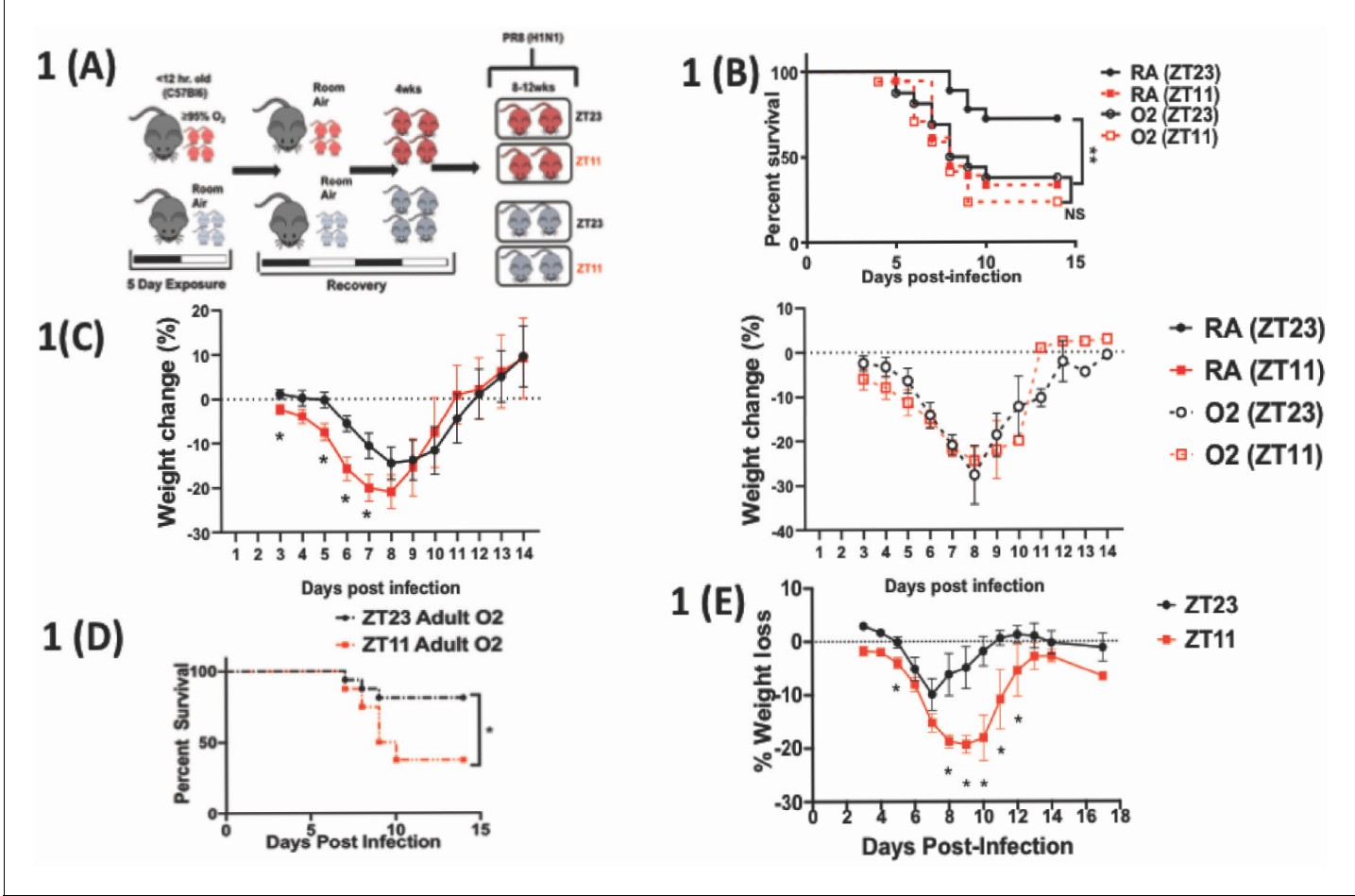

**Figure 1.** Overall experimental design and effects of neonatal hyperoxia on circadian regulation of IAV infection adulthood. (**A**) Experimental model for neonatal hyperoxia followed by IAV infection: C57Bl6 pups aged <12 hr were exposed to either hyperoxia (>95% $O_2$) or room air (21% $O_2$) from PN0–5. Thereafter, the pups were recovered in room air to adulthood (8–12 weeks). As adults, the mice were place in reverse 12 hr light–dark cycles for 2 weeks before being infected with the influenza A virus (PR8). Experimental design: Adult mice were placed in 12 hr light–dark reverse cycles and infected with 25–40 PFU of PR8 at either ZT23 (dawn or just before the onset of rest phase) or at ZT11 (dusk or just before the onset of activity, since mice are nocturnal animals). (**B**) Survival (n = 17–18 per group, **p=0.0027 log-rank test test from three independent experiments) and (**C**) Percentage of body weight lost (n = 10–15 per group, *p<0.001 ANOVA for repeated measures) following IAV infection (25–40 PFU PR8) at either ZT23 (dawn) or at ZT11 (dusk). (**D**) Survival of mice exposed to hyperoxia as adults and infected with IAV after 3–4 weeks of recovery (n = 8–16 per group, *p=0.0417 log-rank test from two independent experiments). (**E**) Percentage of body weight lost (n = 17–18 per group, *p=0.0091 for time of infection by REML mixed effects model for repeated measures) following IAV infection (25–40 PFU PR8) at either ZT23 (dawn or just before the onset of rest phase) or at ZT11 (dusk or just before the onset of activity, since mice are nocturnal animals).

The online version of this article includes the following source data and figure supplement(s) for figure 1:

**Source data 1.** Source data for *Figure 1*.
**Figure supplement 1.** Experimental model for adult hyperoxia followed by IAV infection: C57Bl6/J mice at least 8 weeks old were exposed hyperoxia (>95% $O_2$) for ~48 hr.
**Figure supplement 2.** Body weights post-hyperoxia exposure.

higher than the mice infected at ZT23 (survival 72% in ZT23 vs. 33% in ZT11, p<0.01). By contrast, the mice that had been exposed to hyperoxia as neonates lost this time of day difference: both groups had mortality comparable to the ZT11 (in RA) group (survival in hyperoxia groups: ZT23 at 37% and ZT11 at 25%; *Figure 1b and c*). However, when adult mice exposed to hyperoxia were infected with IAV at either ZT23 or ZT11 (*Figure 1—figure supplement 1*), they maintained the same difference in outcomes as seen in room air controls in *Figure 2b and c*. Thus, exposure to hyperoxia as adults did not appear to impair the circadian control over mortality and morbidity from IAV seen with exposure to neonatal hyperoxia. Considered together, this suggests that early life

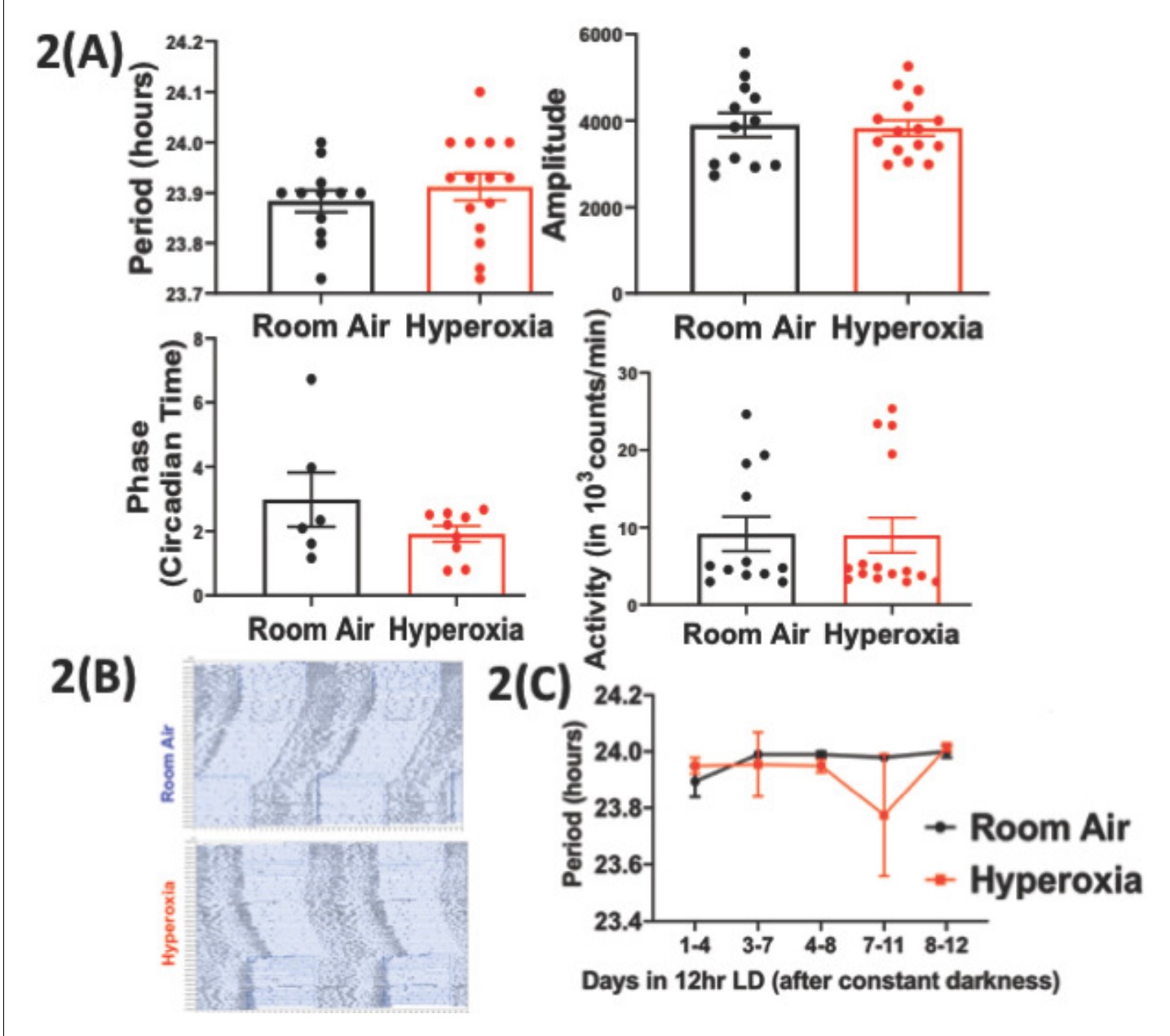

**Figure 2.** Loss of temporal gating of IAV infection in adults exposed to hyperoxia as neonates is not mediated through central circadian clock. Mice pups <12 hr old were exposed to either hyperoxia or room air for 5 days and then recovered in RA into adulthood. As adults, effect on the central (locomotor activity) and peripheral (gene expression from lung and spleen, flow cytometry of lung resident immune cells, and bioluminescence of period two gene) clocks was measured. (A) Central effects were assessed by actigraphy measurements of rest and activity in constant darkness. Period, amplitude, activity counts, and phase of this locomotor activity in adult mice exposed to neonatal hyperoxia or room air (n = 6–15 per group). (B) Representative actigraph images taken from adult mice exposed to either neonatal hyperoxia or room air. This graph depicts the rest–activity behavior from a representative mouse where consecutive days are plotted on the y-axis, and time (in hours) is shown on the x-axis. The black bars represent the number of turns of the running wheel/movement sensed by the infrared motion sensors and indicates a time when the mouse of active. (C) Period length across days to acclimatize from constant darkness (DD) to 12 hr LD conditions adult mice exposed to neonatal hyperoxia or room air (n = 5–6 per group).

The online version of this article includes the following source data for figure 2:

**Source data 1.** Source data for *Figure 2*.

(p0–p5) accords a unique window of influencing circadian regulation and as such exposure to hyperoxia during this period disrupts circadian regulation of the lung injury response to IAV infection.

## Neonatal hyperoxia has only subtle effects on the central clock in adulthood

To determine the effect of neonatal hyperoxia on central circadian rhythms in adulthood, we evaluated the locomotor activity patterns of adult mice (8–12 weeks old of both genders) exposed to 5 days of neonatal hyperoxia ($\geq$95% $O_2$ starting <12 hr of age to 5 days post-natal). A similar duration of hyperoxia is known to cause alveolar oversimplification, as seen in BPD (*Yee et al., 2011*). Immediately after exposure, the pups exposed to hyperoxia weigh less (*Figure 1—figure supplement 2*). However, these pups are indistinguishable from the room air controls in weight or overall activity by 8–12 weeks of age (*Figure 1—figure supplement 2*). To exclude differences in exercise tolerance, we measured locomotor activity using both infrared sensors and running wheels and found that the two methods yielded similar results. Interestingly, while phase, period length, amplitude, and total activity patterns were comparable between the two groups, the hyperoxia group displayed significantly more variation in their period lengths (*Figure 2a*).

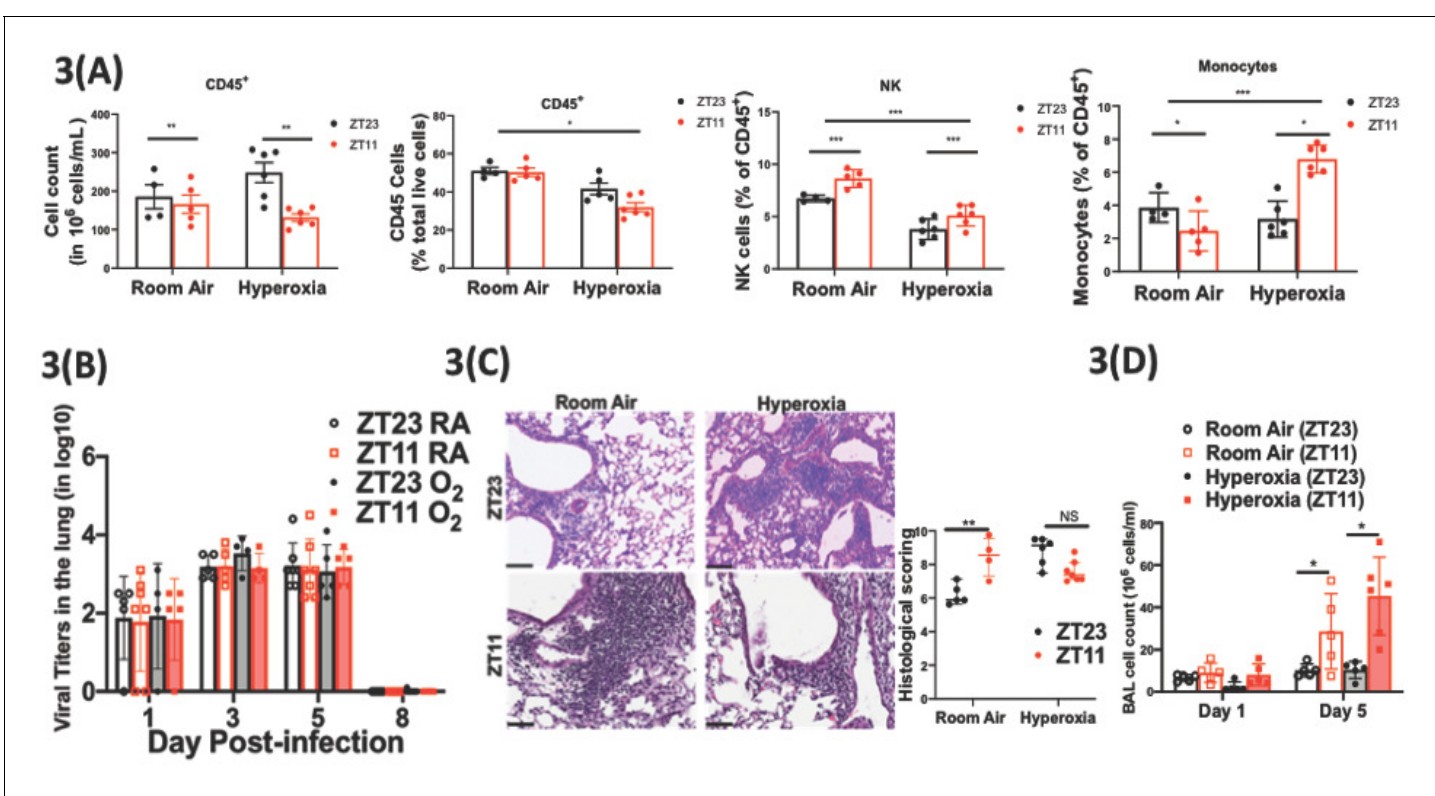

**Figure 3.** Exposure to hyperoxia as neonates has subtle effects of the circadian regulation of the immune response to influenza infection. (A) Flow cytometric enumeration of the CD45$^+$ cells from lungs of adult mice exposed to neonatal hyperoxia or room air (n = 5–7/group from three independent experiments). (B) Viral burden measure by hemagglutination inhibition assay (n = 5–7/group) by two-way ANOVA, p<0.001 for time post-infection, p=0.8855 for treatment group and interaction. (C) Left: Representative micrographs of H and E stained lung sections 8 days after IAV (40 PFU) treatment of adults exposed to neonatal hyperoxia or room air (photomicrograph bar = 100 µm). Right: Severity of lung injury quantified using an objective histopathological scoring system by a researcher blinded to study group (n = 4–8 mice/group; data as median, IQR; Wilcoxon rank sum test; **p<0.01; pooled data from two independent experiments). (D) Bronchoalveolar lavage (BAL) from animals infected at either ZT11 or ZT23 on day 1 and 5 p.i. (n = 5–6/group, p=0.0001 for time at infection, p<0.0001 time after infection, and p=0.0023 for interaction by two-way ANOVA).

The online version of this article includes the following source data and figure supplement(s) for figure 3:

**Source data 1.** Criteria for scoring lung injury on histology.
**Source data 2.** Source data for *Figure 3*.
**Figure supplement 1.** Viral nucleic acid measured by qPCR after IAV infection in adult animals exposed to hyperoxia as neonates (n = 6–8/group).

Infection with IAV disrupts circadian locomotor rhythms under in a mouse model of COPD (*Sundar et al., 2015*), raising the possibility that further stress may unmask a central phenotype. To address the hypothesis that neonatal hyperoxia caused instability of circadian regulation, we examined the ability of the animals to re-entrain to 12 hr light-dark cycles after several weeks in constant darkness. Entrainment refers to the re-synchronization of the internal or endogenous circadian rhythms to external cues (*Golombek and Rosenstein, 2010*), such as the change in lighting schedule here, and is a way to assess the function of the central clock (*Tahara and Shibata, 2018*). Normoxia and hyperoxia groups did not entrain differently to this small change (*Figure 2b and c*). Thus, considered together, these data are consistent with the hypothesis that effects of neonatal hyperoxia on IAV infection are not mediated via lasting changes in the central clock.

## Effect of neonatal hyperoxia on resident immune cells in the lung

We next addressed the possibility that loss of circadian gating of IAV responses in mice exposed to neonatal hyperoxia was secondary to circadian dysregulation of the immune response. We harvested lungs from adult animals exposed to either room air or hyperoxia as neonates and separated different populations of immune cells by flow cytometry at either the beginning of the rest phase (ZT23) or the beginning of the active phase (ZT11). While the total CD45$^+$ population did not differ at these time points in the normoxia group, mice at ZT23 had higher numbers of CD45$^+$ cells than at ZT11 in the hyperoxia group (*Figure 3a*). The percentage of natural killer (NK) was significantly lower in the hyperoxia-exposed mice at both time points. And the phase of the monocytes seems to reverse upon exposure to hyperoxia. Interestingly, in our previous work, more NK cells and lesser inflammatory monocytes were present in the group infected at ZT23 relative to the ZT11 group and thus associated with better outcomes (*Sengupta et al., 2019*). Since these changes in the immune population were noted at baseline, we delved further into the circadian control of immune response to IAV as a mechanism to explain the lack of circadian protection in adults exposed to hyperoxia as neonates.

Immune mechanisms underlying the circadian dysregulation of IAV-induced lung injury in adults exposed to neonatal hyperoxia: We showed previously that circadian control of IAV-induced lung injury and outcomes is not mediated by viral burden, but rather by the control of inflammation, with the group infected at ZT11 revealing evidence of exaggerated inflammation (*Sengupta et al., 2019*). Coupled with our findings above of impaired immune cell regulation in hyperoxia-treated animals, we hypothesized that the loss of time-of-day protection in these animals would be manifest as an exaggerated inflammatory response irrespective of the time of infection.

Consistent with our previous work, we found that the viral burden was comparable between the groups infected at ZT11 or ZT23; this held true in both neonatal hyperoxia- and room-air-exposed groups (*Figure 3b* and *Figure 3—figure supplement 1*). Next, we quantified lung injury on histological analyses using a previously validated scoring system (*Sengupta et al., 2019*). The scores were based on peri-bronchial inflammation, peri-vascular inflammation, alveolar inflammatory exudates, and epithelial necrosis/sloughing. (Scoring system detailed in *Figure 3—source data 2*.) On histological analyses, animals infected in the hyperoxia group scored worse for lung injury and the time of day difference in severity was absent (*Figure 3c*). To address the hypothesis that an exaggerated inflammatory response abrogated circadian variability in the innate immune response of animals exposed to neonatal hyperoxia, we performed bronchoalveolar lavage on days 1 and 5 following infection with IAV. Concordant with our previous work, among control animals exposed to room air as neonates, the room air group showed a higher total BAL count at ZT11 than at ZT23 by day 5 p.i. (*Figure 3d*) among control animals exposed to room air as neonates. Contrary to our hypothesis, this was not influenced by neonatal exposure to hyperoxia, since even among the hyperoxia-exposed pups, the total BAL count was higher in the ZT11 than in the ZT23 group. Thus, impaired circadian regulation of the response to IAV does not appear to arise directly from deficits in the circadian regulation of innate immunity.

## Effect of neonatal hyperoxia on the pulmonary circadian system

We next investigated the effect of hyperoxia on the intrinsic clock in the lung by whole lung gene expression assays as well as with the use of a circadian reporter. On visual inspection, subtle differences in the phase of the oscillation of core clock genes *Bmal1*, *Per2*, and *Nr1d2* were noted as a

consequence of neonatal hyperoxia, but none of these changes were statistically significant (*Figure 4a*). However, since bulk gene expression may conceal functionally relevant changes in cell-specific circadian oscillation, we next used Per2luc mice, exposed them to hyperoxia as neonates and then recorded their bioluminescence ex vivo at 10–11 weeks of age. The hyperoxia-exposed animals showed rhythmic expression of the Period2 gene with comparable periodicity and phase to the animals exposed to room air. However, the amplitude of these oscillations was significantly dampened in the hyperoxia group (*Figure 4b*). These data are consistent with the hypothesis that the perturbation induced by IAV infection unmasks a dysregulation of circadian function intrinsic to the lung in animals exposed to neonatal hyperoxia.

## Disrupting the circadian clock in AT2 cells in adulthood recapitulates the loss of time of day protection seen in adults exposed to neonatal hyperoxia

Neonatal hyperoxia disrupts the development of the respiratory epithelium, at least in part due to the depletion of AT2 cells after recovery (*Yee et al., 2014*). During hyperoxia, the AT2 cell number expands as a source of AT1 cells; upon return to room air for recovery, there is a progressive depletion of AT2 cells, with approximately 70% surviving at 8 weeks of age (*Yee et al., 2014*; *Yee et al.,*

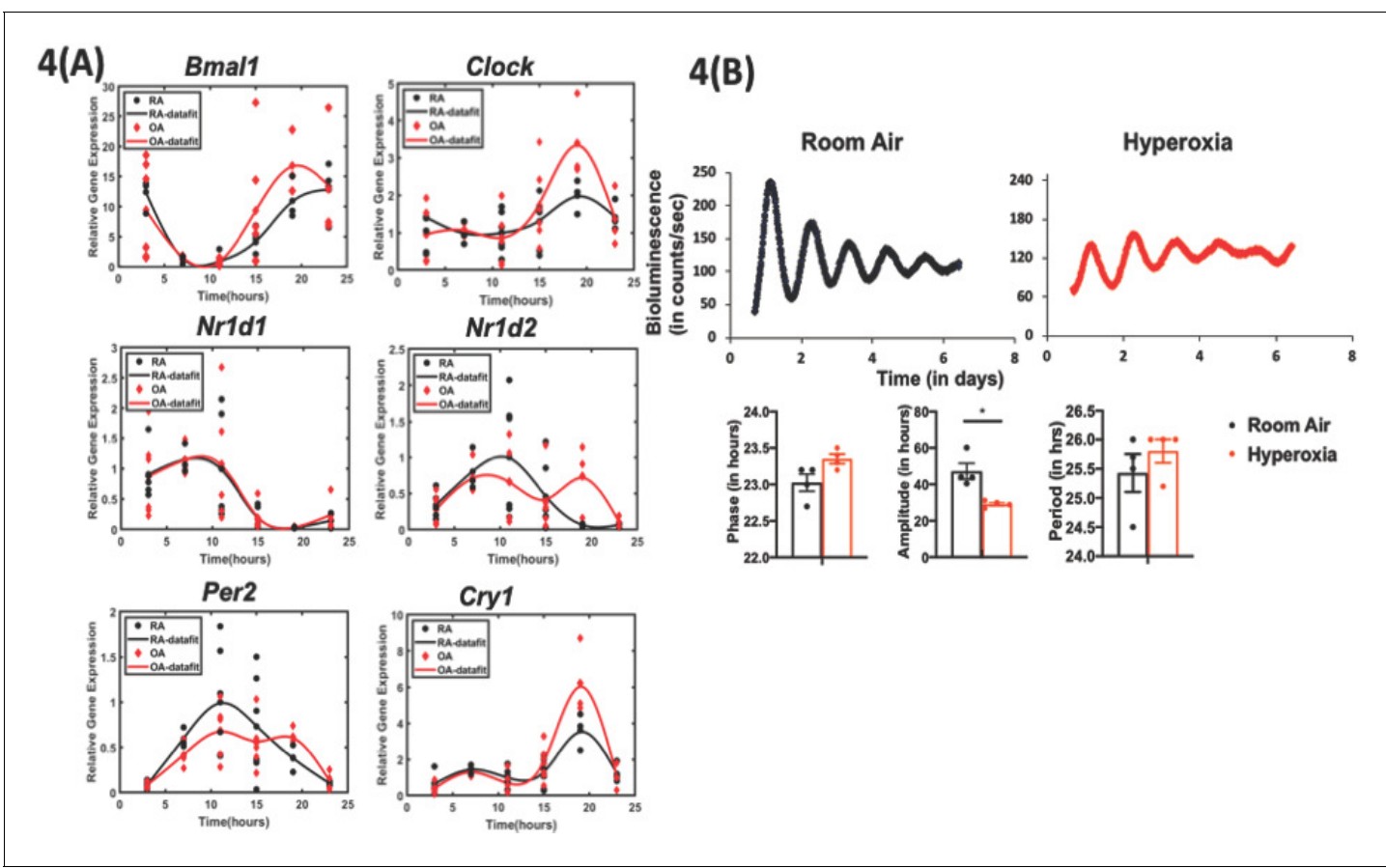

**Figure 4.** Exposure to hyperoxia as neonates reduce the amplitude of circadian oscillations in lung explants from adult animals. (**A**) Gene expression of clock genes from whole lungs harvested at 6 hr intervals determined by qPCR (n = 4–6 per time point from three different experiments). (**B**) Representative bioluminescence tracings, period, amplitude, and phase from lung explants from adult Per2luc mice exposed to neonatal hyperoxia or room air (n = 4 from two independent experiments).

The online version of this article includes the following source data for figure 4:

**Source data 1.** Source data for *Figure 4*.
**Source data 2.** qPCR primers used to generate data for *Figure 4*.

2016). Thus, given the role of the type two airway epithelial cells in hyperoxic injury and repair, we asked if clock disruption in AT2 cells would be sufficient to re-capitulate the effects of neonatal hyperoxia in terms of loss of circadian gating in response to IAV. An effect localized to AT2 cells would also be consistent with the relatively intact circadian gene expression in hyperoxia-treated mice, given that AT2 contribute only a small percentage of the cells in the total lung (*Figure 4a*).

To test the relevance of the clock in AT2 cells, we generated mice in which the core clock gene, *Bmal1* was deleted in these cells (*Sftpc$^{CreERt2/+}$::Bmal1$^{fl/fl}$*). In the tamoxifen-treated *Sftpc$^{CreERt2/+}$:: Bmal1$^{fl/fl}$* mice, *Bmal1* expression was selectively absent in AT2 cells but not elsewhere such as the airway epithelium (*Figure 5—figure supplement 1*). While the cre⁻ mice infected in the morning

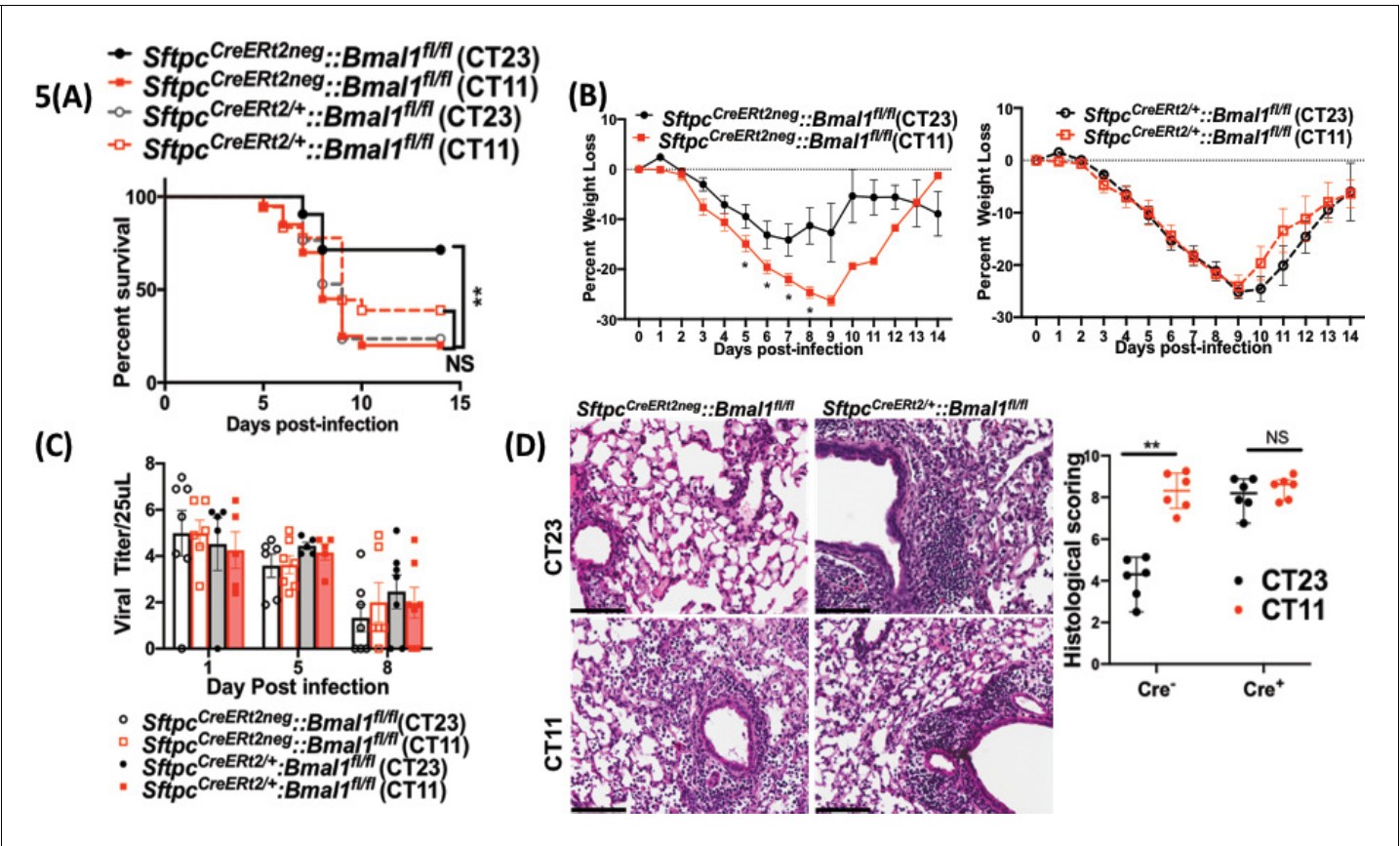

**Figure 5.** Disrupting the circadian clock in AT2 cells in adults recapitulates the phenotype seen in adult animals exposed to hyperoxia as neonates. Experimental design: *Sftpc$^{CreERt2/+}$::Bmal1$^{fl/fl}$* mice (mice lacking *Bmal1* in AT2 cells of the lung epithelium) and their cre$^{neg}$ littermates were treated with tamoxifen at 6–8 weeks of age and acclimatized to reverse cycles of 12 hr LD for 2 weeks. Thereafter, they were maintained in constant darkness for 2–4 days prior to administering IAV (PR8) at CT23 and CT11 and acclimatized to reverse cycles of 12 hr LD for 2 weeks. Thereafter, they were maintained in constant darkness for 2–3 days prior to administering IAV (PR8) at CT23 and CT11. CT23 and CT11 refer to the time corresponding to ZT23 and ZT11, respectively, when animals are maintained under constant darkness conditions. (**A**) Survival curves (n = 16–18 in *Sftpc$^{CreERt2/+}$::Bmal1$^{fl/fl}$* groups and n = 20–21 in cre$^{neg}$, *p=0.0014 Mantel–Cox test; pooled data from four independent experiments). (**B**) Percentage of body weight lost (n = 16–18 in *Sftpc$^{CreERt2/+}$::Bmal1$^{fl/fl}$* groups and n = 18–20 in cre$^{neg}$ group from three independent experiments *p<0.01 ANOVA for repeated measures). (**C**) Viral burden measure by hemagglutination inhibition assay (n = 5–13/group) by two-way ANOVA, p<0.0001 for time post-infection, p=0.9981 for treatment group and interaction. (**D**) Left: Representative micrographs of H and E stained lung sections 8 days after IAV (40 PFU) treatment of *Sftpc$^{CreERt2/+}$:: Bmal1$^{fl/fl}$* mice and their cre⁻ littermates (photomicrograph bar = 100 μm). Right: Severity of lung injury quantified using an objective histopathological scoring system by a researcher blinded to study group (n = 4–8 mice/group; data as median, IQR; Wilcoxon rank sum test; **p=0.0014, CT23 vs. CT11 for Cre⁺ refers to *Sftpc$^{CreERt2/+}$::Bmal1$^{fl/fl}$* mice vs. Cre⁻ which refers to *Sftpc$^{CreERt2neg}$::Bmal1$^{fl/fl}$*; pooled data from three independent experiments).

The online version of this article includes the following source data and figure supplement(s) for figure 5:

Source data 1. Source data for *Figure 5*.

Figure supplement 1. Immunofluorescence staining to demonstrate the AT2-specific loss of Bmal1 in *Sftpc$^{CreERt2/+}$::Bmal1$^{fl/fl}$* mice, but not in *Sftpc$^{CreERt2neg}$::Bmal1$^{fl/fl}$* littermates.

Figure supplement 2. Viral nucleic acid measured by qPCR after IAV infection of *Sftpc$^{CreERt2/+}$::Bmal1$^{fl/fl}$* mice and *Sftpc$^{CreERt2neg}$::Bmal1$^{fl/fl}$* littermates.

had better survival than cre⁻ mice infected in the evening (*Figure 5a*; 72% survival in CT23 group versus 20% survival is CT11 group; p<0.01 by Mental–Cox test), this time of day difference was lost in mice lacking *Bmal1* in their AT2 cells (*Figure 5a*; 25% survival in CT23 group versus 38% in the CT11 group of *Sftpc^CreERt2/+::Bmal1^fl/fl*). CT11 and CT23 refer to the time corresponding to ZT11 and ZT23, respectively, in constant darkness. In models of *Bmal1* deletion, we and others have used constant darkness conditions to avoid any differential effects of light on relevant peripheral clocks (*Gibbs et al., 2014*; *Sengupta et al., 2019*; *Yang et al., 2016*). A pattern similar to the mortality was also noted in the weight loss trajectory from *Sftpc^CreERt2/+::Bmal1^fl/fl* and cre⁻ littermates (*Figure 5b*). Thus, post-natal deletion of *Bmal1* specifically in AT2 epithelial cells thus recapitulated the impact of neonatal hyperoxia on IAV-induced mortality and morbidity.

### *Bmal1* deletion in AT2 cells reiterates the phenotype of circadian dysregulation seen in adult animals exposed to hyperoxia as neonates

As loss of circadian protection may be caused by different mechanisms, we sought to determine whether this loss in the *Sftpc^CreERt2/+::Bmal1^fl/fl* animals was associated with similar underlying mechanisms as those one seen in hyperoxia-exposed animals. Specifically, we determined whether the loss of circadian protection was mediated through anti-viral or host tolerance pathways.

We found that the viral loads at days 1 and 5 and day 8 were comparable across genotypes in animals infected at CT11 or CT23, both via hemagglutination inhibition assay (*Figure 5c*) and viral nucleic acid expression (*Figure 5—figure supplement 1*). This result is identical to that seen in hyperoxia-treated animals exposed to IAV (*Figure 3b*). We next compared the histology of the cre⁺ and the cre⁻ animals infected with IAV and found, again, that it was reminiscent of the hyperoxia-treated animals. While the lung pathology was worse in cre⁻ group infected at CT11 than at CT23, this time of day protection was lost in the cre⁺ animals, where the histology was worse at both CT11 and CT23 (*Figure 5d*). Furthermore, we found that many of the gene expression signatures were comparable between the hyperoxia-treated animals and those with *Bmal1* deleted in AT2 cells. This included a reduction in *Il1b* and *Il10* in both hyperoxia-exposed animals and cre⁺ mice, while the pattern of change was comparable for *ccl2* (all on day 5 p.i. in *Figure 6a*). Finally, even with the loss of *Bmal1* in AT2 cells (*Sftpc^CreERt2/+::Bmal1^fl/fl*), the BAL cell count varied by time of day (*Figure 6b*), being higher in CT11 than at CT23 in both cre+ and cre⁻ animals on day 5 p.i. – again similar to the BAL count patterns seen in adults exposed to hyperoxia. Considered together, these results support the possibility that early-life hyperoxia disrupts the circadian rhythm-mediated protection in IAV-induced lung injury through AT2-clock-mediated effects.

## Discussion

In the current study, we systematically studied the effect of an early-life hyperoxia exposure on the function of circadian rhythms in adulthood. A hallmark of circadian regulation is a difference in outcomes based on time of day at which the insult is sustained. We found that this time of day-specific protection from IAV is lost in adult animals exposed to hyperoxia as neonates, reminiscent of the effect of global genetic disruption of core clock gene *Bmal1* from our previous work (*Sengupta et al., 2019*). Hyperoxia is known to cause significant damage to the lung and one speculation might be that the loss of the time of day difference in outcomes in reflective of a generalized lack of well-being rather than a loss of circadian control. However, our data clearly refutes that possibility, given that the hyperoxia-exposed groups only ever have mortality comparable to the control group infected in the evening; not worse. We have also seen the loss of this circadian protection without worsening beyond the ZT11 group at lower doses of the virus (data not shown).

Consistent with our previous observations on clock disruption, the viral burden after challenge was similar across all the groups. Thus, the loss of protection from IAV in mice exposed to hyperoxia as neonates is not due differences in viral replication. Therefore, we addressed the possibility that this loss of central gating reflected dysregulation of central clock function (measured as locomotor activity rhythms), the immunological response to infection or peripheral clock function in pulmonary epithelial cells.

We found that mice exposure to neonatal hyperoxia did not have a significant effect on the central clock by adulthood. This is not surprising – while other adverse effects early in life are known to disrupt local or central circadian rhythmicity (*Coleman and Canal, 2017*; *Smarr et al., 2017*), the

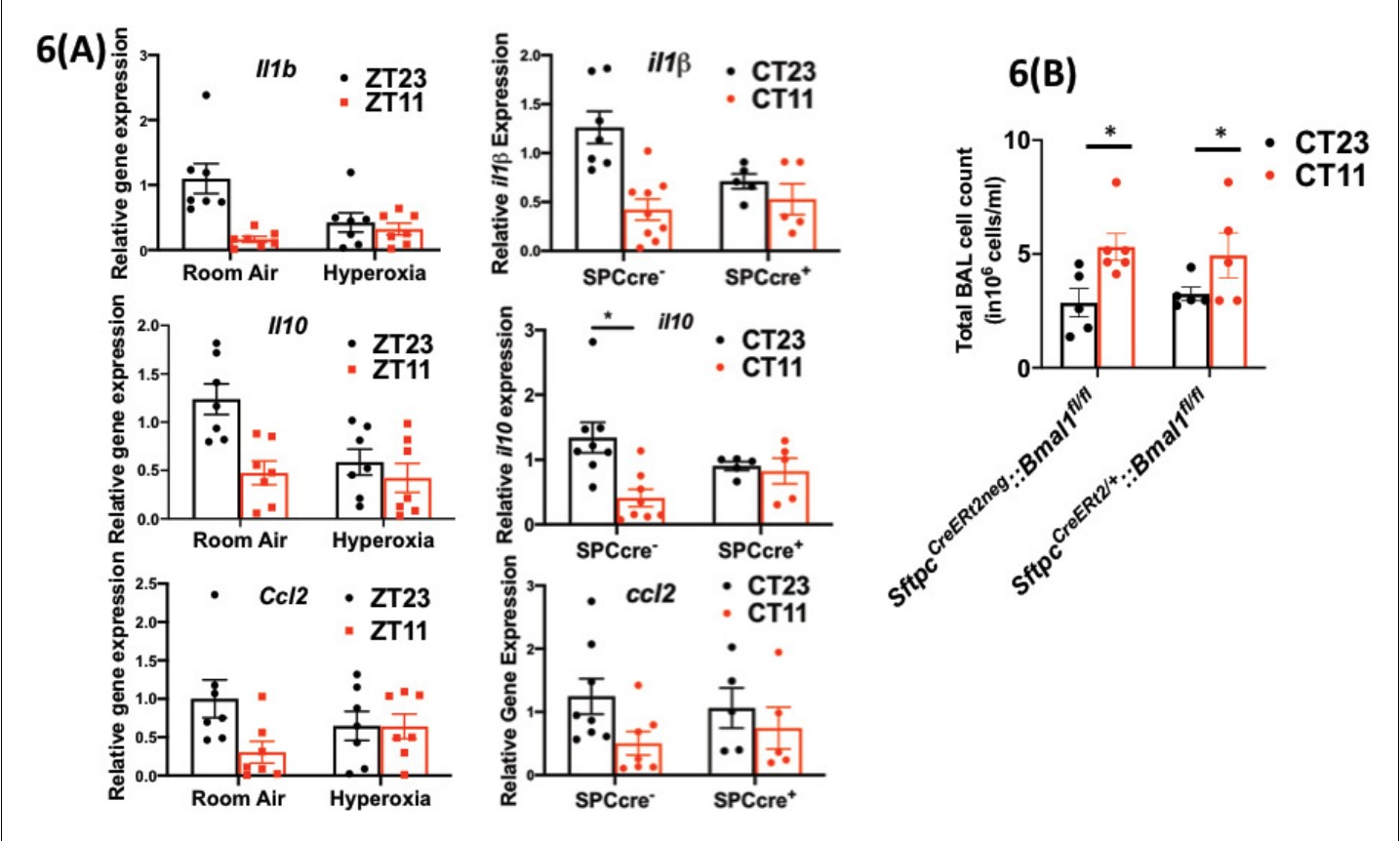

**Figure 6.** Disrupting the circadian clock in AT2 cells in adults recapitulates the gene expression pattern and BAL expression in adult animals exposed to hyperoxia. (**A**) Whole lung gene expression measured by qPCR on day 5 post-infection in adults exposed to neonatal hyperoxia or room air (left panel) and $Sftpc^{CreERt2/+}::Bmal1^{fl/fl}$ mice (mice lacking $Bmal1$ in AT2 cells of the lung epithelium referred to in the figures as SPC-cre[+]) and their $cre^-$ littermates referred to as SPC-cre[−] (right panel). (**B**) Bronchoalveolar lavage (BAL) from animals infected at either CT11 or CT23 on day 5 p.i. (n = 4–6/group, p=0.0063 for time of day by two-way ANOVA).

The online version of this article includes the following source data for figure 6:

**Source data 1.** Source data for *Figure 6*.

SCN clock is well known to be resistant to non-photic stimuli (*Damiola et al., 2000*; *Stokkan et al., 2001*). At baseline, we also saw that neonatal hyperoxia could disrupt the immune clock, which would account for abrogated time of day protection from IAV. Since these cell populations were implicated in the exaggerated inflammation in the groups infected in the evening (*Sengupta et al., 2019*), we initially hypothesized that neonatal hyperoxia disrupts the immune clock which in turn abrogates the circadian protection from IAV. However, following IAV infection total BAL cell counts were higher in the group infected at ZT11 than in those infected at ZT23 in both the room air and neonatal hyperoxia-exposed animals. This is in contrast with the work of O'Reilly et al. who have shown that adults exposed to hyperoxia as neonates had a hyper-inflammatory response, with higher BAL counts in comparison to the room air controls. We speculate that this difference from the work by *O'Reilly et al., 2008* results from our circadian experimental design since we infected mice at dawn and dusk. Furthermore, we found that exposure to hyperoxia did not change the viral burden at either time points, which is consistent with previous literature. Overall, given that even in our previous work, the myeloid clock did not completely control the phenotype seen in animals with a disrupted clock, we focused on the lung epithelium.

The lung consists of anatomically distinct regions (tracheas, bronchioles, and alveoli) each populated by structurally and functionally distinct epithelial cell types. Tracheas and proximal airways are lined by multi-ciliated cells, secretory cells (Scgba1a1[+]), goblet cells, and basal stem/progenitor (BSC) cells. The small airways terminate in the alveolar sacs, which are mainly composed of alveolar

type1 (AT1) or AT2 cells (*Zepp and Morrisey, 2019*). Although all epithelial cells are susceptible to IAV infection, AT2 cells are primarily affected by neonatal hyperoxia. Even though there were only subtle differences in gene expression rhythms assayed in the entire lung, the possibility that the AT2 clock is disproportionately affected cannot be ruled out. We suspect that since AT2 cells contribute only a small portion of the total RNA extracted from the lung, the disruptive effect on the AT2 clock was underestimated. In fact, in our lung explant model, while pulmonary rhythmicity was maintained, the amplitude was definitely blunted in the hyperoxia-treated group. Thus, a role for the AT2 clock is supported by our Bmal1$^{-/-}$ data.

We and others (*Gibbs et al., 2014*; *Sengupta et al., 2019*; *Zhang et al., 2019*) have demonstrated a role for the Scgb1a1 (or CCSP) clock in mediating lung injury, however, this is the first report of a cell intrinsic clock in AT2 cells. Although many clock genes participate in the generation and maintenance of circadian rhythms, *Bmal1* is the only circadian gene whose sole deletion is sufficient to cause arrhythmicity of locomotor activity – the hallmark of circadian disruption. Embryonic deletion of *Bmal1* results in a severe accelerated aging phenotype, in which the circadian phenotype is confounded by potentially off target developmental effects of *Bmal1* deletion (*Yang et al., 2016*). We avoid any confounding by non-circadian effects from *Bmal1* by targeting gene deletion in adulthood. The time of day difference in IAV induced mortality in cre$^-$ littermates was abrogated amongst *Sftpc$^{CreERt2/+}$::Bmal1$^{fl/fl}$* mice, treated with tamoxifen at 8 weeks of age. It is intriguing that the both hyperoxia and AT2 clock disruption not only abolish the circadian directed time of day protection from IAV, but also share many similarities in the mechanisms underlying this dysregulation. In both conditions, the loss of circadian regulation of the host response to IAV is not mediated through control of viral burden, but rather through worse injury. Overall, our data strongly support the possibility that hyperoxia disrupts the circadian regulation of lung injury in IAV through the AT2 clock. Since many of the *Scgb1a1-cre* models affect AT2 cells as well (*Rawlins et al., 2009*), it is possible that the Scgb1a1-cre described previously overestimates the effect of *Bmal1* deletion. Thus, neonatal hyperoxia impairs the circadian regulation of host response to IAV in adult mice.

Our work further strengthens the observation in the field that oxygen tension modulates the circadian clock. While we focused on long-term effects of neonatal hyperoxia on circadian networks, Lagishetty et al. has suggested that hyperoxia exposure in adult mice changes the circadian gene expression, although only a single time point was assessed (*Lagishetty et al., 2014*). Most evidence for the connection between the changes in oxygen tension and circadian clock comes from hypoxia. Interestingly, both clock and HIF belong to the same basic helix-loop-helix PER-ARNT-SIM (bHLH-PAS) transcription factor super family. In the skeletal muscle, a reciprocal relationship was noted between the hypoxia-sensing HIF1 pathways and the circadian clock – the response to hypoxia is gated by the circadian clock and HIF1$\alpha$ binds to clock targets to modulate the clock output under hypoxic conditions (*Peek et al., 2017*). Furthermore, HIF1$\alpha$ is shown to accelerate adaptation to jet-lag, suggesting an interaction between the two at a central level (*Adamovich et al., 2017*). More recently, hypoxia was shown to desynchronize the relationship between peripheral tissue clocks (*Manella et al., 2020*). Our work here suggests that a unique window exists in the lung in the early neonatal period that might affect circadian regulation throughout later life. Since this period, (around p4) affects the development of AT2 population, it was not surprising that AT2-specific disruption of the circadian clock, via *Bmal1* deletion, recapitulated the results from the neonatal hyperoxia cohort. Many developmental pathways such as the Wnt-βcatenin pathways have recently been demonstrated to be central to the pathogenesis of BPD following neonatal hyperoxia (*Sucre et al., 2020*). Thus, one possibility would be that the developmental reprogramming that is triggered by noxious stimuli like neonatal hyperoxia also disrupts the development of the circadian networks within the epithelium. Neonatal hyperoxia affects multiple transcription factors such as *Nfe2l2*, *Cdnk1a*, *Fos*, *Trp53*, *Rela*, *Stat3*, and *Cebpa* (*Wright and Dennery, 2009*). Many of these transcription factors and pathways are known to be under circadian control. Thus, one possibility is that a reciprocal relationship between one or more of these pathways and the circadian clock may underlie the life course effects of hyperoxia on circadian regulation, akin to the *Hif1*a and *Bmal1* connection discussed above. However, since there is not a single regulatory node for hyperoxia unlike the hypoxia–*Hif1*a axis, it is hard to speculate which specific pathway would play this role.

In conclusion, children born prematurely and suffering from even mild BPD have persistent adverse effects on their lung function into adulthood. This may result in part from long-lasting damage to circadian mechanisms residing in AT2 cells that influence the response to injury after viral

infection. The current study provides a circadian paradigm for the long-term consequences of early-life insults in this vulnerable population, paving the way for novel therapeutics and chronobiological strategies.

# Materials and methods

## Key resources table

| Reagent type (species) or resource | Designation | Source or reference | Identifiers | Additional information |
|---|---|---|---|---|
| Antibody | Anti-Bmal1 (Rabbit monoclonal) | Abcam | ab230822 | IF(1:1000) |
| Antibody | anti-Pro-SPC (Rabbit polyclonal) | EMD Millipore | ab3786 | IF(1:2000) |
| Antibody | Nk1.1 (Mouse monoclonal) | Biolegend | 108716 | Flow cytometry (1:100) |
| Antibody | Ly6C (Mouse monoclonal) | Biolegend | 128024 | Flow cytometry (1:100) |
| Antibody | CD45 (Rat monoclonal) | Biolegend | 103114 | Flow cytometry (1:100) |
| Antibody | CD11c (Hamster monoclonal) | Biolegend | 117324 | Flow cytometry (1:100) |
| Antibody | CD11b (Mouse monoclonal) | EBiosciences | 11-0112-41, 47-0112-82 | Flow cytometry (1:100) |
| Antibody | CD3 (Rat monoclonal) | EBiosciences | 11-0032-82 | Flow cytometry (1:50) |
| Antibody | Ly6G (Rat monoclonal) | Biolegend | 127606, 127608, | Flow cytometry (1:100) |
| Antibody | Siglec F (Rat monoclonal) | BD Pharmingen | 552126 | Flow cytometry (1:100) |
| Software Algorithm | Clocklab | Actimetrics | | Rest-activity analyses |
| Other | DAPI stain | | D1306 | (1 µg/mL) |
| Chemicals strain, strain background (include species and sex here) | Tamoxifen | Sigma | T5648-1G | |
| Strain, strain background (include species and sex here) | C57BL/6J | Jax | Stock No: 000664 \| B6 | Both genders and age at infection > 8 weeks |

## Animal models, hyperoxia exposure, and influenza infection

For neonatal hyperoxia, newborn C57BL/6J mice (<12 hr old) were exposed to either 21% oxygen (room air) or $\geq$95% oxygen ($O_2$) between post-natal days 0 and 5. Mice are born in the saccular phase of lung development and thus exposure to hyperoxia shortly after birth simulates the effects of hyperoxia on preterm lungs. This model of BPD is well established. Similar models have resulted in mild alveolar oversimplification and airway hyper-reactivity at 8 weeks of life (*Yee et al., 2011*). To minimize oxygen toxicity, nursing dams were switched out every 24 hr. Following exposure, the hyperoxia-exposed pups were recovered in room air with/alongside their littermates under 12 hr LD conditions until 8–10 weeks of age. For adult hyperoxia exposure, 8 week old C57BL/6J (RRID: IMSR_JAX:000664) mice were exposed to or $\geq$95% oxygen for 48 hr. Thereafter, the mice were recovered in room air for a period of 3–4 weeks and acclimatized to reverse light cycling to split up the group into two for further influenza infection.

For influenza infections, mice were acclimatized to reverse cycles in specially designed circadian chambers for 2 weeks. Thereafter, they were infected at ZT23 or Z11 with 30 PFU of IAV (PR8) i.n. under light isoflurane anesthesia. This method has been used and validated in our previous work and is used by many circadian labs to remove confounding from infecting animals at different times of the conventional day (*Sengupta et al., 2019*).

## Mouse strains and tamoxifen administration

A mouse line with AT2 cell-specific knockout of *Bmal1* was generated by crossing *Sftpc$^{Cre-ERT2/+}$*, a tamoxifen-inducible Cre, with *Bmal1$^{fl/fl}$* mice. The *Sftpc$^{CreERT2/+}$* mice (on mixed Balbc/C567 background) were a kind gift from E.E. Morrissey (University of Pennsylvania; originally made in Dr. H. Chapman's group [*Chapman et al., 2011*] and has been used widely in the field to target AT2 cells). The animals were back crossed for four to five generations onto a C57 background. Tamoxifen

(Sigma–Aldrich) was dissolved/reconstituted in corn oil and ethanol to create a 100 mg/mL stock solution. *Bmal1* deletion was induced by administration of tamoxifen (5 mg dissolved in corn oil/ethanol) via oral gavage for five consecutive days. Following tamoxifen treatment, mice were acclimatized to reverse light–dark cycles for 2 weeks, then placed in constant darkness 2 days before being infected with PR8 (25–35 PFU). Weights were monitored before, during, and post-tamoxifen administration, and no significant weight loss was noted. They were acclimatized to reverse light–dark cycles for 2 weeks and exposed/placed in constant darkness 24–48 days before being infected with PR8 at (CT/ZT23 and 11).

## Flow cytometry

Lungs were harvested after perfusion with 10 mL of phosphate-buffered saline (PBS) through the right ventricle. The lungs were digested using DNAse II (Roche) and Liberase (Roche) at 37°C for 30 min. Dissociated lung tissue was passed through a 70 µm cell strainer, followed by centrifugation and red blood cell (RBC) lysis. Cells were washed and re-suspended in PBS with 2% fetal bovine serum. (Details of the antibodies in Figure 3–Source data 1.) $2–3 \times 10^6$ cells were blocked with 1 µg of anti-CD16/32 (Fc Block) antibody and were stained with indicated antibodies on ice for 20 min. No fixatives were used. Flow cytometric data was acquired using FACS Canto flow cytometer and analyzed using FlowJo software (Tree Star, Inc). All cells were pre-gated on size as singlet live cells. All subsequent gating was on $CD45^+$ in lung only. Neutrophils were identified as live, $CD45^+$, and $Ly6G^+$ cells. $Ly6C^{hi}$ monocytes were identified as live, $CD45^+Ly6G^-Ly6C^{hi}CD11b^+$ cells. NK cells were identified as $CD45^+Ly6G^-LysC^-NK1.1^+CD3^+$ cells.

## Histology and staining

Lungs from the mice were inflated and fixed with 10% neutral buffered formalin (Sigma–Aldrich), embedded in paraffin, and sectioned. Lung sections were stained with hematoxylin–eosin. Stained slides were digitally scanned at 40× magnification using Aperio CS-O slide scanner (Leica Biosystems, Chicago, IL). Representative images were taken from the scanned slides using Aperio Image-Scope v 12.4. Soring was performed blindly using previously validated scoring method (*Sengupta et al., 2019*). Briefly, the eight lung fields selected at random at 20× magnification were scored based on a scale of 0–3 for the following (1) peri-bronchial infiltrates, (2) per-vascular infiltrates, (3) alveolar exudates, and (4) epithelial damage of medium-sized airways. For immunofluorescence staining, paraffin sections of $Sftpc^{CreERt2/+}::Bmal1^{fl/fl+}$ and $Sftpc^{CreERt2neg}::Bmal1^{fl/fl}$ were obtained. Slides were stained with anti-BMAL1 antibody (Abcam, ab230822, 1:1000, RRID:AB_2889035) followed by secondary antibody staining with alexa 488 Straptavdin (A21370, Life Technologies, Eugene, OR, 1:200). Slides were then incubated with anti Pro Surfactant protein (EMD Millipore ab3786, RRID:AB_91588, 1:2000) followed by alexa 594 anti-rabbit (Invitrogen A21207). Slides were then rinsed, counterstained with DAPI, and rinsed again before cover slipping with Prolong Gold (P36930, Life Technologies, Eugene, OR). At least five randomly selected regions were imaged for three samples/genotype on Leica DM 4000 B at 20× and 63×.

## Quantitative PCR

RNA was isolated/extracted from the inferior lobe of the mouse lung using TRIzol (Life Technologies). RNA was further purified using the RNeasy Mini Elute Clean Up Kit (Qiagen). The quantity and quality of RNA were assessed using the NanoDrop ND- 1000 spectrophotometer (NanoDrop Technologies Inc) and cDNA prepared with TaqMan. SYBR Green gene-expression assays were used to measure mRNA levels for genes of interest. Eukaryotic 18S rRNA (Life Technologies) and 28S (Sigma) were used as an internal control for TaqMan and SYBR Green assays, respectively. The samples were run on a Viia7 real-time PCR thermal cycler (Roche), and the relative ratio of the expression of each gene was calculated using the $2^{\Delta\Delta Ct}$ method. Gene expression values for each of *Bmal1*, *Cry1*, *Per2*, *Nr1d1*, and *Nr1d2* were fit by least squares to cosinor curves with a fixed period of 24 hr. Gene expression data was plotted in MATLAB (MATLAB R2020a; The MathWorks Inc) using the curve-fitting function that generates a best-fit line for the given data. A cosinor linear model was fit to the data for each gene (*Cornelissen, 2014*). An F-test comparing the full model allowing for differences in amplitude, phase, and mesor between the hyperoxia and room air conditions to the restricted

model allowing only differences in mesor between conditions yielded no significant results at the p=0.05 level.

Primers: All primers were procured from Thermofisher life sciences. Catalog numbers are provided in source data file.

## Viral titration

As described before (*Sengupta et al., 2019*), we harvested lungs at different time points following infection, as indicated in the specific experiment. Following homogenization in PBS–gelatin (0.1%), these lungs were frozen for preservation. The presence of influenza virus was evaluated using MCDK (RRID:CVCL_0422) cells with 1:10 dilutions of the lung homogenates at 37°C. After 1 hr of infections, 175 µL of media containing 2 µg/mL trypsin was added and the cells were further incubated for 72 hr at 37°C. A total of 50 µL of medium was then removed from the plate and tested by hemagglutination of chicken RBCs for the presence of virus particles. The hemagglutination of RBCs indicated the presence of the virus.

## Rest–activity measurements

Mice were singly housed in circadian light control boxes (Actimetrics) to measure their locomotor or rest–activity rhythms using either running wheels or Infrared Motion Sensors (Actimetrics). After the mice were acclimatized to circadian chambers, their activity was measured in 12 hr LD conditions for 2 weeks, constant dark (DD) conditions for 2–4 weeks, and again in 12 hr LD conditions. Period, amplitude, phase, and total activity counts were calculated. In the second shift from constant darkness to 12 hr LD conditions, the rate of entrainment to the new lighting system was also analyzed. We used the period length averaged over days 1–4, 3–7, 4–8, 7–11, and 8–12 was determined for each group. Data was analyzed using Clocklab software.

## Ex vivo bioluminescence recording from lung explants

Lungs from Per2luc mice were harvested following perfusion to flush out blood cells. Thereafter, peripheral sections were obtained, and explants were placed in media containing luciferin. Bioluminescence recording was initiated, and data was recorded for 5–6 days. Data was analyzed using Clocklab software.

## Statistical analyses

Sample size analyses was performed using PS Sample size calculator. For survival experiments, sample size of 15 was calculated for median survival times (5 in ZT11 and 28 in ZT23, although the experiments were often plotted for 14 days) and an alpha of 0.05 with 90% power, matching 1:1 for control to case. Data was platted and analyzed using Graphpad Prism Software. Graphs represent data as means ± SEM or median ± IQR as applicable. Based on the distribution of data, either parametric (Student's t-test or two-way ANOVA based on number or type of comparison groups) or non-parametric (Mann–Whitney or Kruskal–Wallis test) was used. Mortality data was analyzed using Mantel–Cox test. Tukey corrections were used for multiple comparisons.

## Statement on rigor and reproducibility

All studies were performed using animals from either Jackson Labs and animals from in-house breeding. The background strain of each genetically modified animal has been specified, and controls were cre⁻ littermates on that same background. Reported findings are summarized results from 3 to 6 independent experiments.

## Approval

All studies involving IAV infection were carried out in biosafety level 2 (BSL2 and ABSL2) conditions and approved facility. Animals were provided with food and water ad libitum. All procedures and studies were performed by experienced personnel with appropriate supervision from veterinary support staff. All animal studies were approved by the University of Pennsylvania animal care and use Committee (protocol ID # 805645) and met the stipulations of the Guidelines for the Care and Use of Laboratory Animals.

## Acknowledgements

We are thankful to members of the FitzGerald lab – Dr. S Teegarden for help with animal breeding and general lab management; Dr. G Grant for his comments and discussions at lab meetings. This work was supported by the NHLBI-K08HL132053 (SS), NICHD-K12HD043245 (SS), a Maturational Human Biology grant from the Institute of Translational Medicine and Therapeutics, University of Pennsylvania (SS), and NIH/NCRR RR023567 (GAF). Dr. FitzGerald is the McNeil Professor of Translational Medicine and Therapeutics and a senior advisor to Calico Laboratories. We are also thankful to Dr. Carolina B Lopez for her advice regarding viral stock generation and viral titration.

## Additional information

### Competing interests

Amita Sehgal: Reviewing editor, *eLife*. The other authors declare that no competing interests exist.

### Funding

| Funder | Grant reference number | Author |
| --- | --- | --- |
| National Heart, Lung, and Blood Institute | K08HL132053 | Shaon Sengupta |
| National Institute of Child Health and Human Development | K12HD043245 | Shaon Sengupta |
| National Center for Research Resources | RR023567 | Garret A FitzGerald |
| Howard Hughes Medical Institute | | Amita Sehgal |
| National Center for Advancing Translational Sciences | 5T32MH106442-04 | Thomas G Brooks |
| National Center for Advancing Translational Sciences | NCATS-5UL1TR000003 | Thomas G Brooks |

The funders had no role in study design, data collection and interpretation, or the decision to submit the work for publication.

### Author contributions

Yasmine Issah, Formal analysis, Investigation, Methodology, Writing - original draft; Amruta Naik, Investigation, Methodology, Project administration, Writing - review and editing; Soon Y Tang, Kaitlyn Forrest, Mara Mermigos, Investigation, Methodology; Thomas G Brooks, Formal analysis, Writing - review and editing; Nicholas Lahens, Formal analysis, Visualization; Katherine N Theken, Conceptualization, Writing - review and editing; Amita Sehgal, Conceptualization, Resources, Visualization, Writing - review and editing; George S Worthen, Conceptualization, Visualization, Writing - review and editing; Garret A FitzGerald, Conceptualization, Resources, Software, Funding acquisition, Writing - review and editing; Shaon Sengupta, Conceptualization, Resources, Data curation, Supervision, Funding acquisition, Validation, Investigation, Visualization, Methodology, Project administration, Writing - review and editing

### Author ORCIDs

Nicholas Lahens ![ORCID] https://orcid.org/0000-0002-3965-5624
Katherine N Theken ![ORCID] http://orcid.org/0000-0001-9918-9170
Shaon Sengupta ![ORCID] https://orcid.org/0000-0001-5237-3835

### Ethics

Animal experimentation: All animal studies were approved by the University of Pennsylvania Animal Care and Use Committee and adhered strictly of the Guide for the care and Use of Laboratory

animals. No survival surgeries were involved. Any potential pain and suffering was minimized by using a priori generated and validated guidelines (protocol ID # 805645).

## Decision letter and Author response
Decision letter https://doi.org/10.7554/eLife.61241.sa1
Author response https://doi.org/10.7554/eLife.61241.sa2

## Additional files

### Supplementary files
• Transparent reporting form

### Data availability
All data generated or analysed during this study are included in the manuscript and supporting files. Source data files have been provided for all figures.

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
