## [Decision Letter]

**Acceptance summary:**

Your study shows a clear role for neonatal hyperoxia to perturb circadian pathways that regulate respiratory responses to influenza infection that is likely to have wide-ranging effects for other respiratory pathogens.

**Decision letter after peer review:**

Thank you for submitting your article "Loss of circadian Protection against Influenza Infection in Adults Exposed to Hyperoxia as neonates" for consideration by *eLife*. Your article has been reviewed by two peer reviewers, one of whom is a member of our Board of Reviewing Editors, and the evaluation has been overseen by Miles Davenport as the Senior Editor. The reviewers have opted to remain anonymous.

The reviewers have discussed the reviews with one another and the Reviewing Editor has drafted this decision to help you prepare a revised submission.

Essential revisions:

1) Edit paper to address reviewer #1 points 1-3 and 5-8 listed below to increase the readability of the for a broad readership.

2) To edit the paper to suggest a mechanism for the proposed effects of hyperoxia on circadian signalling pathways in the lung and impact of other respiratory pathogens. The authors should cite a recent BioRixv pre-print: Yee et al., 2020.

3) To provide the following additional data sets to strengthen their overall conclusions:

– Provide additional viral replication parameters beyond PCR enumeration of influenza viral RNA such as measuring viral infectivity or viral antigen immunostaining in the lung

– Include data to validate Bmal1 gene KO in AT2 cells – such as protein staining in the KO cells.

– Address stage specificity of hyperoxia on circadian gene expression

Reviewer #1:

Issah and colleagues show that neonatal hyperoxia perturbs circadian protection from influenza infection in a mouse model that is phenocopied by a genetic deletion of Bmal1 in alveolar type 2 (AT2) epithelial cells. These elegant experiments provide a mechanism for why neonatal hyperoxia associates with increased susceptibility to influenza associated disease in adult life. The manuscript is well written and overall the experimental data support the stated conclusions, however, both the text and figure legends are brief and further information would increase the readability of the paper. Encourage the authors to discuss the translational impact of their results for treating IAV infection and other viruses that infect the respiratory tract such as SARS-CoV-2.

1) The brevity of the manuscript text may reflect the authors drafting of the paper for a general audience, resulting in text that could be read as “superficial” by chronobiologists or viral immunologists. For example: “The mechanism involves both the pulmonary epithelium and the immune system.” “Changes in oxygen tension are known to affect the clock as well as the interplay between the SCN and peripheral clocks.” I would encourage the authors to expand and provide mechanistic data for some these statements.

2) Current Figure 1 and Figure 2 could be combined into a single figure.

3) Figure 3B – can the authors provide more details on what this figure shows bearing in the mind the general readership of the journal.

4) Figure 4B. The authors should provide more detail on how they measure viral burden and why these time points were selected. Have they evaluated infectious virus levels rather than simple PCR quantification of IAV RNA. This is the one area of the study that I felt was over-interpreted and further data could substantiate their conclusions. The colors used for the bar charts in this figure are not easy to discern – similar criticism of Figure 4D. Suggest the authors recolor the figures and provide an overlay of all data points.

5) Figure 4C – it is not clear what the data points in the histological scoring represent and why they differ between ZT11 and ZT23.

6) Figure 5A was poorly described with limited stat analysis and very low resolution figures. The accompanying text states: “the cycling of Cry1 and Clock was slightly increased” is not appropriate and the authors should address this figure.

7) Figure 6 – revise choice of colors and symbols to help clarify these figures.

8) Figure 7A – provide an overlay of data symbols for all bar charts.

Reviewer #3:

This well-written manuscript by Issah et al. identifies a new potential mechanism by which injury in early life may contribute to lung disease in adulthood. Using a previously established influenza infection model and AT2-cell specific CRE knockout of clock gene Bmal1, the authors show evidence that saccular stage hyperoxia injury disrupts the lung cellular clocks, resulting in a loss of the protective mechanism from influenza injury later in life. While the manuscript is clearly written and the proposed mechanism is novel, there are a few additional experiments necessary to strengthen the findings of this paper:

1) All of the data shown for Bmal expression after injury is at the level of RNA by qPCR. If possible, showing AT2-specific loss of Bmal by RNA in situ hybridization or at the protein level by immunofluorescence (with AT2 co-localization) would be very helpful.

2) Is the effect of hyperoxia on clock gene expression stage specific? Data are shown for neonatal/saccular stage injury--what happens to the lung cellular clocks of mice exposed to hyperoxia after P5 or in adulthood? These are important controls to consider.

3) It would be helpful for the authors to propose a mechanism in the Discussion whereby hyperoxia leads to cellular clock disruption and to link this proposed mechanism to other relevant papers discussing aspects of the molecular mechanisms of saccular stage hyperoxia injury.

---

## [Author Response]

Essential revisions:1) Edit paper to address reviewer# 1 points 1-3 and 5-8 listed below to increase the readability of the for a broad readership.

We have addressed the points below.

2) To edit the paper to suggest a mechanism for the proposed effects of hyperoxia on circadian signalling pathways in the lung and impact of other respiratory pathogens. The authors should cite a recent BioRixv pre-print: Yee et al., 2020.

Thank you for drawing our attention to this report. The preprint is now published in Scientific reports and we have included this reference in our manuscript.

3) To provide the following additional data sets to strengthen their overall conclusions:– Provide additional viral replication parameters beyond PCR enumeration of influenza viral RNA such as measuring viral infectivity or viral antigen immunostaining in the lung

We have processed the samples for hemagglutination inhibition assay for both the hyperoxia cohort as well as the AT2 Bmal1KO animals.

– Include data to validate Bmal1 gene KO in AT2 cells – such as protein staining in the KO cells.

We have included IF-staining with Bmal1 and AT2 marker, Pro-SPC in lung sections (in Figure 5—figure supplement 1).

– Address stage specificity of hyperoxia on circadian gene expression

We have included data for adults exposed to hyperoxia and infected with influenza virus, after a period of recovery. (Figure 1D and E).

Reviewer #1:Issah and colleagues show that neonatal hyperoxia perturbs circadian protection from influenza infection in a mouse model that is phenocopied by a genetic deletion of Bmal1 in alveolar type 2 (AT2) epithelial cells. These elegant experiments provide a mechanism for why neonatal hyperoxia associates with increased susceptibility to influenza associated disease in adult life. The manuscript is well written and overall the experimental data support the stated conclusions, however, both the text and figure legends are brief and further information would increase the readability of the paper. Encourage the authors to discuss the translational impact of their results for treating IAV infection and other viruses that infect the respiratory tract such as SARS-CoV-2.1) The brevity of the manuscript text may reflect the authors drafting of the paper for a general audience, resulting in text that could be read as “superficial” by chronobiologists or viral immunologists. For example: “The mechanism involves both the pulmonary epithelium and the immune system.” “Changes in oxygen tension are known to affect the clock as well as the interplay between the SCN and peripheral clocks.” I would encourage the authors to expand and provide mechanistic data for some these statements.

We thank the reviewer for this insight and have accordingly expanded the sections as recommended in not only the above two instances but others where the critique would be relevant.

Upon further review, the comment on epithelium and immune cells did not seem to be pertinent here, so have deleted that reference altogether.

Revised as below:

“Changes in oxygen tension are known to affect the clock (Walton et al., 2018; Wu et al., 2017). […] More recently, even a short burst of hypoxia was shown to desynchronize the relationship between the SCN and peripheral clocks (Manella et al., 2020), which would confer further risk under conditions of stress.”

2) Current Figure 1 and Figure 2 could be combined into a single figure.

Done.

3) Figure 3B – can the authors provide more details on what this figure shows bearing in the mind the general readership of the journal.

We have clarified the same in both the Materials and methods section and the legends that accompany the figure. Revised section as below:

Main text: “To address the hypothesis that neonatal hyperoxia caused instability of circadian regulation, we examined the ability of the animals to re-entrain to 12hr light-dark cycles after several weeks in constant darkness. […] Normoxia and hyperoxia groups did not entrain differently to this small change [Figure 2B and C]”.

Materials and methods: “Mice were singly housed in circadian light control boxes (Actimetrics) to measure their locomotor or rest-activity rhythms using either running wheels or Infrared Motion Sensors (Actimetrics). […] Data was analyzed using Clocklab software.”

Legends for new “Figure 2B” (was earlier Figure 3B): “Representative actigraph images taken from adult mice exposed to either neonatal hyperoxia and room air. […] The black bars represent the number of turns of the running wheel/movement sensed by the infrared motion sensors and indicates a time when the mouse of active.”

4) Figure 4B. The authors should provide more detail on how they measure viral burden and why these time points were selected. Have they evaluated infectious virus levels rather than simple PCR quantification of IAV RNA. This is the one area of the study that I felt was over-interpreted and further data could substantiate their conclusions. The colors used for the bar charts in this figure are not easy to discern – similar criticism of Figure 4D. Suggest the authors recolor the figures and provide an overlay of all data points.

We thank the reviewer for this comment. We have expanded our dataset for both eth hyperoxia and the Sftpc cohorts. For hyperoxia animals we now have days 1, 3, 5 and 8 and for the Sftpc-cre we have days 1, 5 and 8. Our aims was to capture the early replication and clearance trajectory following IAV infection. Further, we have expanded our viral burden determination from qPCR to include the hemagglutination inhibition assay. We have included the results from Hemagglutination inhibition assay for the hyperoxia and the SPC-cre animals. We are glad to note that overall, the conclusion of the two methods of viral burden determination are the same. Time of day at infection doesn’t impact viral replication/burden in room air or neonatal hyperoxia exposed animals or for the Sftpc status.

Figure 4D: We have also changed the color scheme of the figure and overlaid the individual data points on top of the summary stats.

5) Figure 4C – it is not clear what the data points in the histological scoring represent and why they differ between ZT11 and ZT23.

To clarify this response we have included the scoring scheme as a table in the source data (Figure 3—source data 2) and expanded on the same in the text. The data points in the histological scoring is the summary from these measurements. Revised as:

“Next, we quantified lung injury on histological analyses using a previously validated scoring system(Sengupta et al., 2019). […] On histological analyses, animals infected in the hyperoxia group scored worse for lung injury and the time of day difference in severity was absent [Figure 3C].”

6) Figure 5A was poorly described with limited stat analysis and very low resolution figures. The accompanying text states: “the cycling of Cry1 and Clock was slightly increased” is not appropriate and the authors should address this figure.

We have revised the Materials and methods and the Results section pertaining to this experiment, as recommended by the reviewer. Further, we have also revised the figures with better resolution and overall clarity. Revised text as:

“On visual inspection, subtle differences in the phase of the oscillation of core clock genes *Bmal1*, *Per2* and *Nr1d2* were noted as a consequence of neonatal hyperoxia, but none of these changes were statistically significant [Figure 4A].”

7) Figure 6 – revise choice of colors and symbols to help clarify these figures.

We have updated the colors and legends.

8) Figure 7A – provide an overlay of data symbols for all bar charts.

Done.

Reviewer #3:This well-written manuscript by Issah et al. identifies a new potential mechanism by which injury in early life may contribute to lung disease in adulthood. Using a previously established influenza infection model and AT2-cell specific CRE knockout of clock gene Bmal1, the authors show evidence that saccular stage hyperoxia injury disrupts the lung cellular clocks, resulting in a loss of the protective mechanism from influenza injury later in life. While the manuscript is clearly written and the proposed mechanism is novel, there are a few additional experiments necessary to strengthen the findings of this paper:1) All of the data shown for Bmal expression after injury is at the level of RNA by qPCR. If possible, showing AT2-specific loss of Bmal by RNA in situ hybridization or at the protein level by immunofluorescence (with AT2 co-localization) would be very helpful.

We have performed immunofluorescence with Bmal1 and Pro-SPC on lung sections from Sftpc-cre+Bmal1^fl/fl^ as well as Sfptc-cre+Bmal1^+/+^ littermates--results confirming an AT2 specific *Bmal1* deletion in the former and are included as Figure 5—figure supplement 1.

2) Is the effect of hyperoxia on clock gene expression stage specific? Data are shown for neonatal/saccular stage injury--what happens to the lung cellular clocks of mice exposed to hyperoxia after P5 or in adulthood? These are important controls to consider.

We thank the reviewer for this fascinating line of inquiry. We have likewise repeated the hyperoxia exposure in adults followed by IAV infection after a recovery period of 3-4 weeks. It appears that the effect of hyperoxia on the circadian control of IAV induced outcomes is specific to the saccular stage. The experimental scheme is detailed in Figure 1—figure supplement 1 and the results in Figure 1D and E.

3) It would be helpful for the authors to propose a mechanism in the Discussion whereby hyperoxia leads to cellular clock disruption and to link this proposed mechanism to other relevant papers discussing aspects of the molecular mechanisms of saccular stage hyperoxia injury.

We have included an entire section discussing the possible interconnections as proposed by the reviewers. Revised section as below:

“Our work further strengthens the observation in the field that oxygen tension modulates the circadian clock. […] However, since there is not a single regulatory node for hyperoxia unlike the hypoxia-HIF1α axis, it is hard to speculate which specific pathway would play this role.”